# Assessing and predicting adolescent and early adulthood common mental disorders using electronic primary care data: analysis of a prospective cohort study (ALSPAC) in Southwest England

Daniel Smith [1,2] Kathryn Willan [3] Stephanie L Prady [4]
Josie Dickerson [3] Gillian Santorelli [3] Kate Tilling [1,2]
Rosie Peggy Cornish [1,2]

[1]MRC Integrative Epidemiology Unit, University of Bristol, Bristol, UK
[2]Population Health Sciences, Bristol Medical School, University of Bristol, Bristol, UK
[3]Born in Bradford, Bradford Teaching Hospitals NHS Foundation Trust, Bradford, UK
[4]Department of Health Sciences, University of York, York, UK

**Correspondence to**
Dr Daniel Smith;
dan.smith@bristol.ac.uk

## ABSTRACT

**Objectives** We aimed to examine agreement between common mental disorders (CMDs) from primary care records and repeated CMD questionnaire data from ALSPAC (the Avon Longitudinal Study of Parents and Children) over adolescence and young adulthood, explore factors affecting CMD identification in primary care records, and construct models predicting ALSPAC-derived CMDs using only primary care data.

**Design and setting** Prospective cohort study (ALSPAC) in Southwest England with linkage to electronic primary care records.

**Participants** Primary care records were extracted for 11 807 participants (80% of 14 731 eligible). Between 31% (3633; age 15/16) and 11% (1298; age 21/22) of participants had both primary care and ALSPAC CMD data.

**Outcome measures** ALSPAC outcome measures were diagnoses of suspected depression and/or CMDs. Primary care outcome measure were Read codes for diagnosis, symptoms and treatment of depression/CMDs. For each time point, sensitivities and specificities for primary care CMD diagnoses were calculated for predicting ALSPAC-derived measures of CMDs, and the factors associated with identification of primary care-based CMDs in those with suspected ALSPAC-derived CMDs explored. Lasso (least absolute selection and shrinkage operator) models were used at each time point to predict ALSPAC-derived CMDs using only primary care data, with internal validation by randomly splitting data into 60% training and 40% validation samples.

**Results** Sensitivities for primary care diagnoses were low for CMDs (range: 3.5%–19.1%) and depression (range: 1.6%–34.0%), while specificities were high (nearly all >95%). The strongest predictors of identification in the primary care data for those with ALSPAC-derived CMDs were symptom severity indices. The lasso models had relatively low prediction rates, especially in the validation sample (deviance ratio range: −1.3 to 12.6%), but improved with age.

**Conclusions** Primary care data underestimate CMDs compared to population-based studies. Improving general practitioner identification, and using free-text or secondary

## Strengths and limitations of this study

► We used a large prospective cohort (Avon Longitudinal Study of Parents and Children; ALSPAC) with validated mental health questionnaires to assess depression and common mental disorders (CMDs) (which we treat as our 'reference standard') and were able to link these data to individuals' electronic primary care records, with this linkage data covering ~80% of the cohort.

► We were able to assess agreement between ALSPAC data and electronic primary care data for CMDs across adolescence and into adulthood, a key life transition and period where mental health problems often emerge.

► There is a risk of selection bias, as many participants with primary care data did not have ALSPAC mental health measures, while primary care data coverage also decreased with age; continued participation in both cases is likely to be non-random.

► For this study we assumed that the CMD measures from ALSPAC are the 'reference standard' against which the primary care data should be compared; however, these measures may also be subject to misclassification.

► The available linkage data consisted of primary care Read codes, which misses data from other clinical sources, such as secondary care or from primary care free-text data.

care data, is needed to improve the accuracy of models using clinical data.

## INTRODUCTION

Common mental disorders (CMDs; depression and anxiety) are a leading cause of morbidity, disability and premature death worldwide.[1] Rates of CMDs have increased over the past few decades,[2] including in adolescence and early adulthood,[3] where

these conditions frequently first appear.[4 5] The prevalence of CMDs in childhood and adolescence (age 5–16) in the UK is estimated to be 4%,[6] rising to 16% among 16–24 years old[7]; these can have significant long-term consequences, including on education, quality-of-life, employment and physical and mental health.[5 8 9]

Assessing the prevalence of CMDs in the population, especially in adolescence, is essential for monitoring, research and planning appropriate public health services. Estimates of prevalence could be from population studies (which are expensive and time-consuming to conduct), or using primary (general practitioner; GP) and secondary (hospital and specialised healthcare services) care records.[10–13] However, CMDs are often underdiagnosed in routine primary care data (the so-called 'clinical iceberg' phenomenon), with over half of all depressed patients with clinical symptoms of depression not recognised as such.[14 15] Reasons for this include: individuals with CMDs not visiting their GP[16]; GPs misdiagnosing, or being reticent in diagnosing, CMDs[15]; and GPs increasingly recording symptoms, rather than specific diagnoses.[17] This 'clinical iceberg' may be particularly prevalent among children and adolescents, who may be less likely to visit their GP. Additionally, GPs may fail to identify, or be less willing to diagnose CMDs or prescribe medication to these groups, potentially in part due to a lack of confidence of GPs in identifying and managing CMDs in children and adolescents, low mental health literacy and help-seeking behaviour among these groups, and GP visits being too brief to elicit such sensitive information.[18–22] Primary care physicians frequently refer to secondary care services, such as Child and Adolescent Mental Health Services (CAMHS[23]), again contributing to the under-reporting of adolescent CMDs in primary care records.

To assess the accuracy of primary care-derived CMD rates, these must be compared against a reference standard.[16] A systematic review in adults found that, relative to a reference standard, specificity is generally high (few false positives) but sensitivity is rather low (many false negatives[12]). Previous research from the Avon Longitudinal Study of Parents and Children (ALSPAC) compared linked primary care records at age 17/18 against CMDs measured on 1562 participants via the revised Clinical Interview Schedule (CIS-R).[10] Using CIS-R as the reference standard, this study found that—similar to findings in adults—sensitivities were low while specificities were high. Together, these findings suggest that primary care data may significantly underestimate the prevalence of CMDs in the population.

Previous UK research has shown that greater symptom severity is the strongest predictor of attending primary care regarding mental health.[16] Other factors, such as age, sex and employment status, also predicted access to primary care, but their contributions were weaker.[16] In contrast, a smaller US study of individuals with depressive symptoms found no demographic differences between those who sought help and those who did not, although symptom severity again predicted help-seeking behaviour.[24] Socio-demographic factors may play a role in access to primary care, recognition of symptoms and access to treatment, which contribute to continuing health inequalities.[25 26] For instance, a UK study found that both ethnicities other than white British and low socioeconomic position (SEP) predicted lower rates of CMD detection in primary care records during the maternal period.[27] Even if individuals with a CMD do contact a physician, the likelihood of receiving treatment is also dependent on symptom severity, as well as socio-demographic factors.[28 29]

Models predicting 'true' CMD status from variables available in primary care records could help to identify the prevalence of individuals with 'missing' CMDs as well as the factors predicting these cases. Previous work has predicted CMDs based on an Australian dataset,[30] but did not use primary care records, so its utility may be limited as some relevant factors are unlikely to be present in routine health records (eg, job satisfaction, social isolation, being a carer, having a partner, etc). Research using only primary care record data to predict validated measures of CMDs from population-based studies is therefore required.

This study has three aims:

1. Replicate and expand the results of a previous ALSPAC study at age 17/18 (~2800 participants[10]) by including additional participants with linkage data (~12 000 participants[31 32]), and explore agreement between primary care records and cohort data across multiple time points over adolescence and young adulthood (ages 15–23).
2. Assess the factors impacting rates of CMD identification in primary care records.
3. Construct prediction models, with ALSPAC-measured CMDs as the outcome, to predict CMD status using only primary care data.

## METHODS
### Study design and participants

ALSPAC is a pregnancy-based longitudinal birth cohort which recruited pregnant women in the Bristol area of Southwest England with an expected delivery date between 1 April 1991 and 31 December 1992.[33 34] A total of 14 541 eligible pregnancies were initially recruited into the study, with a total of 14 676 fetuses, resulting in 14 062 live births, of which 13 988 were alive at 1 year of age. After further waves of post-natal recruitment, as of February 2019, there are a total of 14 901 study child participants enrolled in ALSPAC who were alive at 1 year.[32] These children and their parents have been followed since birth, with detailed data collected via questionnaires, in-person clinic assessments, and linkage to routine data sets. The study website contains details of all available data through a fully searchable data dictionary and variable search tool: http://www.bristolac.uk/alspac/researchers/our-data/. From 2014 onwards (when the study children were aged 22 years) data were collected and managed using REDCap

electronic data capture tools hosted at the University of Bristol.[35] We used the Strengthening the Reporting of Observational Studies in Epidemiology cohort reporting guidelines.[36]

When the study children reached legal adulthood (age 18), ALSPAC initiated a postal fair processing campaign to formally re-enrol the children into the study (prior to this parent-based consent was mandatory, although from age 9 children assented to data collection as well) and to simultaneously seek opt-out permission for ALSPAC to link to their health and administrative records.[37] Linkage to primary care records was carried out following this campaign and electronic primary care records have been extracted for nearly 12 000 study children.[32] This linkage is described in more detail in online supplemental material (see also Cornish *et al*[31]).

In total, 14 731 ALSPAC participants were eligible for our study, comprising all enrolled singletons and twins who were alive at 1 year of age and had not withdrawn consent from the study (data from triplet/quadruplet births were excluded due to confidentiality reasons, as per standard ALSPAC protocols). Of this total sample, 13 113 participants were sent fair processing materials, of which 368 (2.8%) dissented to linkage. Primary care records (although not necessarily for the entire time period) were extracted for 11 807 of these individuals (80% of the original 14 731 eligible participants; 90% of the 13 113 sent fair processing materials). Note that there are several dynamic factors that affect inclusion eligibility in these analyses (eg, study enrolment status and linkage quality to the National Health Service (NHS) Person Demographics Service). Therefore, the numbers reported here may differ from the numbers reported in the ALSPAC primary care linkage data note (currently in preparation).

The current study includes ALSPAC data from multiple time points between the ages of 15 and 23 (table 1), from either clinic or questionnaire data collections. The age 15/16 and 17/18 clinics collected data on both depression and anxiety; at the other time points only depression was assessed. Linked primary care record data coverage decreases with age because the linkage data primarily covers the Bristol area; as many participants moved away as they reached adulthood (eg, for university or work) they are lost from the linked dataset.

## ALSPAC data

At the age 15/16 clinic, depression and anxiety were assessed using the Development and Well-Being Assessment (DAWBA) interview,[38] which identifies several psychiatric diagnoses in children and adolescents (based on International Classification of Diseases-10 (ICD-10) and Diagnostic and Statistical Manual of Mental Disorders fourth edition (DSM-IV) criteria). Here, in addition to a diagnosis for depression by itself, we defined CMDs as a diagnosis of depression and/or any anxiety disorder (generalised anxiety disorder, panic disorder, agoraphobia, social phobia and specific phobias).

At the 17/18 clinic, depression and anxiety were assessed using a self-administered computerised CIS-R questionnaire.[39] As with DAWBA, CIS-R can be used to assign ICD-10 diagnoses of depression and anxiety disorders.[40] Here, anyone diagnosed with mild, moderate or severe depression was classified as having depression, while a diagnosis of CMD was defined having depression and/or an anxiety disorder (generalised anxiety disorder, mixed anxiety and depression, panic disorders and phobic disorders).

At the other ages (16/17, 18/19, 21/22 and 22/23 questionnaires), depression was assessed using a self-administered Short Mood and Feelings Questionnaire (SMFQ), a 13-item questionnaire assessing depressive symptoms over the past 2 weeks.[41] Total SMFQ scores range between 0 and 26, with a score of 12 or more frequently used as a diagnosis of depression.[42] Although there are problems of inaccuracy with using cut-offs from questionnaires as screening tools for depression,[43] using ALSPAC data the validity of the SMFQ during childhood and adolescence was found to be high when compared against ICD-10-derived depression diagnoses from CIS-R at age 17/18.[44] Only participants who answered all 13 SMFQ questions were included in the analyses.

To compare sociodemographic differences between those with and without linked primary care data and to

**Table 1** Details of ALSPAC data used and coverage with primary care linkage data

| Age (time point) | Measure | With ALSPAC CMD data (n) | With primary care data n (%) |
|---|---|---|---|
| Age 15/16 (TF3 clinic) | DAWBA (depression and anxiety) | 5332 | 3663 (68.7) |
| Age 16/17 (CCS questionnaire) | SMFQ (depression only) | 4950 | 3213 (64.9) |
| Age 17/18 (TF4 clinic) | CIS-R (depression and anxiety) | 4534 | 3084 (68.0) |
| Age 18/19 (CCT questionnaire) | SMFQ (depression only) | 3302 | 1982 (60.0) |
| Age 21/22 (YPA questionnaire) | SMFQ (depression only) | 3283 | 1298 (39.5) |
| Age 22/23 (YPB questionnaire) | SMFQ (depression only) | 3896 | 1325 (34.0) |

CIS-R, Clinical Interview Schedule-revised; DAWBA, Development and Well-Being Assessment; SMFQ, Short Mood and Feelings Questionnaire.

**Table 2** Details of the multiple definitions of 'depression' and 'CMD' derived from the primary care data

| Definition | Description |
| --- | --- |
| Current diagnosis | Current diagnosis of depression/CMD |
| Current diagnosis, treated | Current diagnosis of depression/CMD and currently receiving treatment |
| Current diagnosis or treatment or symptoms | Current diagnosis or symptoms or treatment for depression/CMD |

CMD, common mental disorder.

explore whether demographic factors impact rates of identification in primary care records, several variables measured during pregnancy or at birth and known to be predictive of non-response in ALSPAC were used.[33 34] These include child sex; maternal age, home ownership status; marital status and parity; parental education levels and child ethnicity. Additional variables used for aims 2 and 3 are discussed below.

### Electronic primary care data

The linked primary care data comprised Read codes V.2 (5 bytes), along with associated dates. Read codes are a comprehensive list of standardised clinical terms used by healthcare professionals within the UK NHS to record clinical information (they have since been replaced by 'SNOMED CT' codes, but our data contained Read codes as they predated this change). Read codes relevant to diagnosis, symptoms or treatment (antidepressants, anxiolytics and hypnotics) of depression or anxiety (including phobic disorders) were extracted.[10 11] These were combined to produce three definitions of depression and CMDs (table 2). Based on previous research,[10] these were chosen to include the definitions with the lowest sensitivity ('current diagnosis, treated'), the highest sensitivity ('current diagnosis or treatment or symptoms'), and an intermediate sensitivity which is also the most straightforward to extract from primary care records ('current diagnosis'). 'Current' diagnoses, symptoms or treatment were defined as being 6 months either side of the age the study child attended the clinic or completed the questionnaire and 'historical' as having occurred at least 6 months prior to the age at the clinic/questionnaire. Note that treatment does not include psychological therapies, even though these are the recommended first line of treatment for adolescents, as these therapies are mainly given by specialist secondary mental health services and may not be noted in primary care records. Read codes were used to identify referrals to mental health services. A list of the Read codes used is provided in online supplemental table 1.

Additional data were extracted to predict identification in primary care records and for the prediction models. These primary care variables may be associated with our outcomes of interest, and included: average annual number of GP consultations and prescriptions at the relevant time point; current and historical somatic and general symptoms (defined in online supplemental table 1); referral to mental health services; common chronic health conditions (asthma and eczema); other mental health conditions (eating disorders, attention deficit hyperactivity disorder (ADHD), conduct disorder, autistic spectrum disorder, alcohol and drug abuse, schizophrenia, bipolar disorder and psychosis); family and personal history of depression and mental health issues; self-harm and smoking status (described in more detail below).

To ensure that only individuals with primary care data at the relevant time points were included, inclusion criteria were: (a) having associated linkage data; (b) having primary care data for at least 6 months after their clinic visit or questionnaire completion (based on GP registration dates) and (c) first appearing in the primary care records a minimum of 18 months prior to their clinic visit or questionnaire completion (allowing a 6-month window for 'current' data, plus a whole year previous for 'historical' data).

### Statistical analysis

For each primary care definition and at each time point, sensitivity, specificity and positive and negative predictive values (NPVs) were calculated separately for predicting ALSPAC-derived measures of depression and CMDs (if measured). Exact 95% CIs were derived using binomial probabilities.

We then explored factors associated with identification of CMDs/depression in primary care records for individuals diagnosed in the ALSPAC data. As primary care diagnosis numbers were low, we used the definition with the highest sensitivity ('current diagnosis or symptoms or treatment'). Univariate logistic regression was used to explore whether each covariate was associated with identification. The variables used to predict identification were a combination of ALSPAC and primary care data (for a full list see online supplemental table 2). These identification analyses were repeated for each timepoint, separately for both depression and CMD (if measured).

Finally, lasso (least absolute selection and shrinkage operator) models were used at each time point to assess the combination of variables from primary care data which best predicted ALSPAC-derived depression/CMDs. Lasso models apply a lambda weight which constrains weakly predictive variables falling below this value to zero, while also shrinking remaining non-zero coefficients towards zero. This results in sparse models which minimise overfitting, and can subsequently be used for out-of-sample prediction.[45 46] Tenfold cross-validation was used for all lasso models and visual inspection of the cross-validation

plots were performed to assess that the optimal lambda value had been selected.

We randomly split our sample into 60% training and 40% validation samples, and then compared the deviance ratios for each to inspect how well the training model performed when predicting depression/CMDs in the validation sample. The deviance ratio is a goodness-of-fit statistic generally between 0 and 1 comparable to $R^2$, but for non-linear models, indicating the fraction of deviance explained relative to the null constant-only model (with values closer to 1 indicating better model fit (Hastie et al., p33)[45]). In-sample deviance ratios refer to results from the 60% training sample, while out-of-sample deviance ratios refer to results from the 40% validation sample (note that as the coefficients derived from the training sample are used in the validation sample, it is possible to observe negative deviance ratios, indicating that the coefficients are *worse* at predicting the outcome than the null model). Logistic lasso models were used with ALSPAC-derived depression or CMDs at each time point as the outcome variable, and all variables in online supplemental table 3 as predictors.

To assess whether these models, which use all the available information held in primary care records, increase model fit relative to just the primary care diagnosis/symptoms/treatment data, we compared these models against: (a) a prediction model which just contained 'current diagnosis' as a predictor variable and (b) a prediction model which included 'current diagnosis', 'current symptoms' and 'current treatment' as predictor variables. In-sample and out-of-sample deviance ratios of these models were compared with assess model fit.

For each time point, from the models using all the available primary care data we also calculated the predicted probabilities of receiving a depression or CMD diagnosis (with a threshold of >50% probability to define diagnosis) in the 40% validation sample, and compared sensitivities and specificities derived from these prediction models against the three definitions using the raw primary care data (table 2). All analyses were conducted using Stata V.16.0.

### Patient and public involvement statement

ALSPAC has an advisory panel of >30 participants who meet bimonthly to advise on study design, methodology and acceptability. Investigations were carried out into participants' understanding of and feelings towards the acceptability of linking to health and administrative data and, in particular, to access sensitive health information such as data on mental and sexual health. This information was obtained through a qualitative study and through discussions with ALSPAC's this advisory panel. However, the advisory panel were not involved in the specific study reported here. ALSPAC communicates with participants via regular newsletters and has an active website and social media presence.

## RESULTS

### Demographics and linkage data coverage

Table 1 shows numbers with both linkage and ALSPAC data at each time point (the reasons individuals who have ALSPAC data, but do not have linkage data, are provided in online supplemental table 4). The proportion of unlinked records increases with age, most likely because these individuals left the area as they became adults.

Comparisons between those with ALSPAC data who do and do not have primary care data are presented in table 3 (for age 15/16 and 22/23 time points) and online supplemental table 5 (for all other time points). There are some differences, particularly in terms of socio-economic position (eg, less likely to have primary care data if higher parental education levels), but little difference in terms of sex. At later time points, participants with more GCSEs (General Certificate of Secondary Education; mandatory UK qualifications taken at age 16) or equivalents are less likely to have primary care data. Few differences in depression/CMD diagnosis are apparent between these two groups. With the exception of CMD/depression diagnoses (which increases with age), differences in demographics across the time points are minimal, although the proportion of females with ALSPAC data does increase over time.

Figure 1 gives the proportions with a current diagnosis of depression/CMD in the primary care data comparing those who did to those who did not complete the ALSPAC clinic or questionnaire. Those with ALSPAC data are more likely to have a current CMD diagnosis, particularly at the later time points. For depression, those with ALSPAC data are slightly more likely to have a primary care diagnosis at ages 21/22 and 22/23 but there are no differences at earlier time points.

### Sensitivity, specificity and predictive values

We focused first on the age 17/18 clinic data (table 4), the results of which were broadly consistent with previous ALSPAC analyses.[10] At this age, 243/3084 participants (7.9%) were diagnosed as depressed using the ALSPAC CIS-R data, while 455 (14.8%) met the criteria for ALSPAC diagnosis of CMD. Using the various primary care definitions, the number of individuals diagnosed as depressed ranged from a minimum of 20 (0.7%) using a definition of 'current diagnosis, treated', to a maximum of 122 (4%) using a definition of 'current diagnosis or symptoms or treatment'. Thus, the sensitivity of each primary care definition to predict ALSPAC-derived CMDs was low, ranging from 3.7% to 24.3%. Specificity was higher (all >97.8%), as was the NPVs (all >92%), while positive predictive values (PPVs) ranged between 45% and 58.4%. For the primary care CMD data, numbers diagnosed ranged from a minimum of 29 (0.9%) to a maximum of 171 (5.5%). Sensitivity (range: 3.5%–19.1%) and specificity (all >96.8%) to predict ALSPAC-derived CMDs were marginally lower compared with depression at this age, while each PPV was slightly higher (range: 47.4%–55.2%) and NPV lower (range: 85.6%–87.4%).

**Table 3** Demographics at the age 15/16 clinic and 22/23 questionnaire time points

| | Age 15/16 (TF3 clinic)—DAWBA (CMDs and depression) | | Age 22/23 (YPB questionnaire)—SMFQ (depression) | |
| --- | --- | --- | --- | --- |
| | Primary care data (n=3663) | No primary care data (n=1669) | Primary care data (n=1325) | No primary care data (n=2571) |
| **Sex** | | | | |
| Male | 1748 (47.7%) | 782 (46.9%) | 474 (35.8%) | 868 (33.8%) |
| Female | 1915 (52.3%) | 887 (53.2%) | 851 (64.2%) | 1703 (66.2%) |
| Maternal age at child's birth | 29.2 (4.6) | 29.3 (4.6) | 29.2 (4.5) | 29.6 (4.4) |
| **Mother's home ownership status** | | | | |
| Owned/mortgaged | 2878 (85.3%) | 1292 (83.8%) | 998 (84.2%) | 2022 (85.5%) |
| Rented | 137 (4.1%) | 90 (5.8%) | 58 (4.9%) | 121 (5.1%) |
| Council/housing association | 281 (8.3%) | 111 (7.2%) | 100 (8.4%) | 151 (6.4%) |
| Other | 79 (2.3%) | 48 (3.1%) | 30 (2.5%) | 70 (3%) |
| **Mother's marital status** | | | | |
| Never married | 460 (13.5%) | 211 (13.6%) | 152 (12.6%) | 262 (11%) |
| Single/divorced | 147 (4.3%) | 83 (5.3%) | 50 (4.2%) | 105 (4.4%) |
| First marriage | 2607 (76.5%) | 1149 (74.0%) | 918 (76.2%) | 1851 (77.7%) |
| 2nd/3rd marriage | 194 (5.7%) | 110 (7.1%) | 85 (7.1%) | 164 (6.9%) |
| **Mother's parity** | | | | |
| 0 | 1623 (48.1%) | 787 (51.3%) | 560 (47.0%) | 1155 (49.1%) |
| 1 | 1201 (35.6%) | 507 (33.1%) | 413 (34.7%) | 815 (34.7%) |
| 2 or more | 554 (16.4%) | 240 (15.7%) | 218 (18.3%) | 381 (16.2%) |
| **Mother's highest education level** | | | | |
| O level/lower | 1884 (55.2%) | 765 (49.3%) | 748 (62.8%) | 1071 (45.5%) |
| A level | 920 (27.3%) | 476 (30.7%) | 280 (23.5%) | 770 (30.6%) |
| Degree | 565 (16.8%) | 312 (20.1%) | 164 (13.8%) | 562 (23.9%) |
| **Father's highest education level** | | | | |
| O level/lower | 1428 (46.2%) | 559 (39.3%) | 550 (50.5%) | 795 (35.9%) |
| A level | 947 (30.6%) | 437 (30.6%) | 348 (32%) | 653 (29.5%) |
| Degree | 716 (23.2%) | 429 (30.1%) | 191 (17.5%) | 767 (34.6%) |
| **Child ethnicity** | | | | |
| White | 3179 (95.9%) | 1466 (95.8%) | 1138 (96.8%) | 2237 (96.2%) |
| Non-white | 137 (4.1%) | 65 (4.3%) | 38 (3.2%) | 88 (3.8%) |
| GCSEs (n) (or equivalents) | 7.3 (3.6) | 7.5 (3.6) | 7.3 (3.5) | 8.3 (3.2) |
| **ALSPAC depression diagnosis** | | | | |
| No | 3606 (98.4%) | 1638 (98.1%) | 1103 (83.2%) | 2180 (84.6%) |
| Yes | 57 (1.6%) | 31 (1.9%) | 223 (16.8%) | 398 (14.5%) |
| **ALSPAC common mental disorder (CMD) diagnosis** | | | | |
| No | 3540 (96.6%) | 1622 (97.2%) | – | – |
| Yes | 123 (3.4%) | 47 (2.8%) | – | – |

Of those with ALSPAC data, the table compares those who have primary care data against those who do not. For categorical variables, cells are counts and percentages. For continuous variables, cells are means and SD. Note also that the denominators vary as the variables come from different data sources, with different levels of completeness. As the demographics are broadly similar across all time points, only the first and last time point are presented here (see online supplemental table 5 for all other time points).
ALSPAC, Avon Longitudinal Study of Parents and Children; CMDs, common mental disorders; DAWBA, Development and Well-Being Assessment; GCSE, General Certificate of Secondary Education; SMFQ, Short Mood and Feelings Questionnaire.

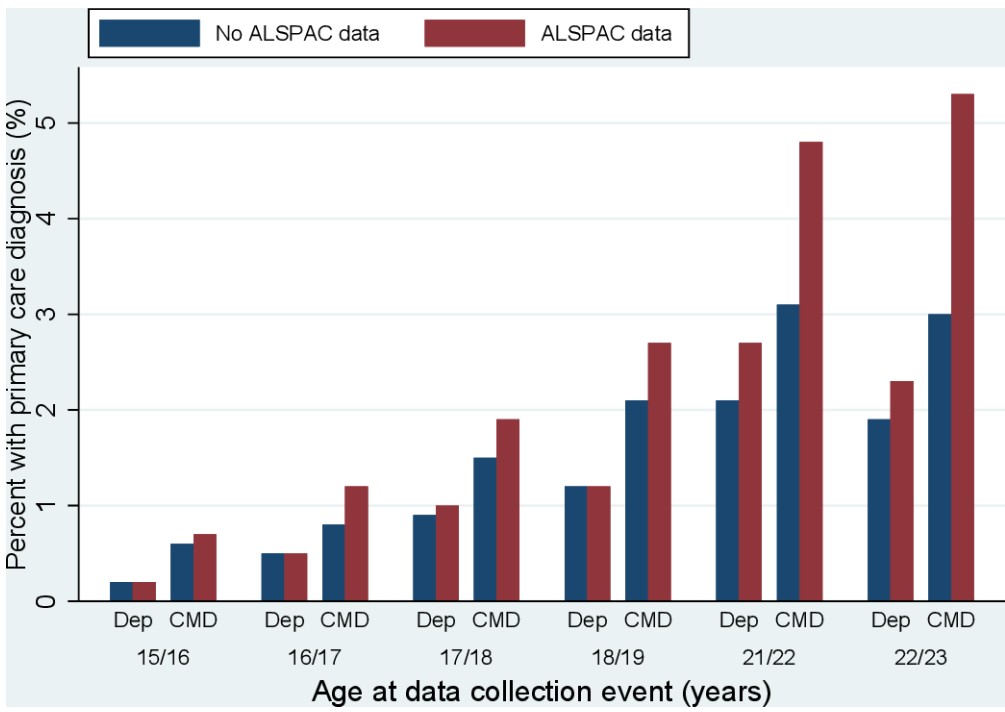

**Figure 1** Comparing primary care common mental disorder (CMD) and depression rates in participants with versus without Avon Longitudinal Study of Parents and Children (ALSPAC) data at each time point. For participants who did not attend the clinic/complete the questionnaire, the age to define a 'current' diagnosis was based on ±6 months from the average age each clinic/questionnaire was completed. Individuals who have primary care data and attended/completed the clinic/questionnaire, but do not have ALSPAC-derived depression/CMD data (as this session was not completed for whatever reason), are not included in the figure. Full details of these numbers, and the data for this figure, are provided in online supplemental table 6.

Similar results were found for the age 15/16 clinic using the DAWBA measure (online supplemental table 7), although with fewer depression and CMD diagnoses (in both primary care and ALSPAC) and lower sensitivities. The comparison between ALSPAC SMFQ questionnaire data and primary care data at ages 16/17, 18/19, 21/22 and 22/23 are displayed in online supplemental tables 8–11, and were similar to those using DAWBA (age 15/16 clinic) and CIS-R (age 17/18 clinic), with relatively high specificity but low sensitivity for all primary care definitions of depression when predicting ALSPAC-derived depression. Sensitivity increased with age, while specificity decreased (figure 2).

### Identification of CMDs/depression cases in primary care records

Next, we explored the factors associated with identification as a case from the primary care data in those with an ALSPAC-derived diagnosis. Results are presented in full in online supplemental table 12 (giving ORs and 95% CIs for all analyses) and online supplemental figure 1 (providing a graphical summary of key results over each time point). There are few consistent associations of sociodemographic factors (parental education, child sex, child education, etc) with being identified as a case in the primary care records. Primary care case identification was more likely in participants with greater symptom severity. Some primary care covariates (eg, smoking status, eating disorder and other mental health issues) were associated

with higher rates of primary care case identification at younger ages, but had weaker associations at later ages. Others (somatic and general symptoms, higher consultation/prescription rates, referrals to mental health services and self-harm status) were consistently associated with higher rates of primary care case identification. Due to the low numbers diagnosed as having CMD or depression at the age 15/16 clinic, both from the DAWBA assessment and from primary care records, results from this time point should be treated with caution.

### Predicting ALSPAC CMDs/depression from primary care records

The in-sample deviance ratios, fitted on the 60% training sample, and the out-of-sample deviance ratios, fitted on the 40% validation sample, for each time point are displayed in table 5. In general, in-sample deviance ratios are quite low (8.3%–14.6%). Out-of-sample deviance ratios are lower (−1.3 to 12.6%) but do increase with age. The penalised coefficients from these prediction models are presented in online supplemental table 13, with full models to estimate predicted probabilities given in online supplemental table 14. Many factors from the primary care data consistently predicted ALSPAC CMD/depression diagnoses across many time points, including: being female, a history of self-harm, number of GP consultations, referral to mental health services and historical and/or current depression diagnoses/symptoms/treatment. Several associations were time point specific,

**Table 4** Depression and CMD diagnoses based on the CIS-R (Clinical Interview Schedule-revised) data from the age 17/18 TF4 clinic against various definitions derived from the primary care data at this age (n=3084)

| | | CIS-R depression | | | CIS-R CMD | | |
|---|---|---|---|---|---|---|---|
| Primary care definition | | No | Yes | Total | No | Yes | Total |
| Current diagnosis | No | 2825 | 229 | 3054 | 2599 | 428 | 3027 |
| | Yes | 16 | 14 | 30 | 30 | 27 | 57 |
| Sensitivity | | 5.8% (3.2–9.5) | | | 5.9% (3.9–8.5) | | |
| Specificity | | 99.4% (99.1–99.7) | | | 98.9% (98.4–99.2) | | |
| Positive predictive value | | 46.7% (28.3–65.7) | | | 47.4% (34.0–61.0) | | |
| Negative predictive value | | 92.5% (91.5–93.4) | | | 85.9% (84.6–87.1) | | |
| | | No | Yes | Total | No | Yes | Total |
| Current diagnosis, treated | No | 2830 | 234 | 3064 | 2616 | 439 | 3055 |
| | Yes | 11 | 9 | 20 | 13 | 16 | 29 |
| Sensitivity | | 3.7% (1.7–6.9) | | | 3.5% (2.0–5.6) | | |
| Specificity | | 99.6% (99.3–99.8) | | | 99.5% (99.2–99.7) | | |
| Positive predictive value | | 45% (23.1–68.5) | | | 55.2% (35.7–73.6) | | |
| Negative predictive value | | 92.4% (91.4–93.3) | | | 85.6% (84.3–86.9) | | |
| | | No | Yes | Total | No | Yes | Total |
| Current diagnosis or symptoms or treatment | No | 2778 | 184 | 2962 | 2545 | 368 | 2913 |
| | Yes | 63 | 59 | 122 | 84 | 87 | 171 |
| Sensitivity | | 24.3% (19.0–30.2) | | | 19.1% (15.6–23.0) | | |
| Specificity | | 97.8% (97.2–98.3) | | | 96.8% (96.1–97.4) | | |
| Positive predictive value | | 48.4% (39.2–57.6) | | | 50.9% (43.1–58.6) | | |
| Negative predictive value | | 93.8% (92.9–94.6) | | | 87.4% (86.1–88.6) | | |

This table also includes sensitivities, specificities, positive predictive values (PPV) and negative predictive values (NPV) for the depression and common mental disorder (CMD) diagnoses based on the CIS-R data from this clinic (with 95% confidence intervals displayed in brackets). In these analyses we are treating the Avon Longitudinal Study of Parents and Children data as the reference standard.

occurring in only one or two models (eg, smoking at TF4 depression and CCS, eczema for TF4 depression, conduct disorder at CCS, etc). These coefficients should not be interpreted causally, especially is there is high collinearity between variables (as is likely to be present here given that many variables measure similar constructs).

For all time points other than age 15/16 clinic depression, the 'full' prediction model (based on the set of all primary care variables; online supplemental table 3) performed better than both the 'diagnosis only' and 'diagnosis/symptoms/ treatment' models for both in-sample and out-of-sample prediction (online supplemental table 15).

Sensitivities from these prediction models were marginally higher than for definitions of 'current diagnosis' and 'current diagnosis with treatment', but lower than the 'current diagnosis or treatment or symptoms' sensitivities. However, the specificities of the prediction models were greater than the 'current diagnosis or treatment or symptoms' definition, and on par with the stricter definitions based on 'current diagnosis' or 'current diagnosis with treatment' (online supplemental table 16).

These prediction models therefore appear to more accurately detect cases of depression/CMD compared with these more stringent definitions from the primary care records, while also avoiding many of the false negatives associated with broader definitions (such as 'current diagnosis or treatment of symptoms'). However, sensitivities from these prediction models are still rather low, ranging between 3.5% and 16.3% (all specificities are >98%).

## DISCUSSION

This study compared primary care data against validated measures of CMDs at multiple time points during adolescence and young adulthood. Taking ALSPAC data as the reference standard, our results demonstrate that, regardless of how CMDs are defined from primary care records, sensitivities are low across all ages (range: 1.6%–34%). However, detection of CMDs in primary care records does improve with age. Specificities were high, with most above 95%. This suggests that the primary care data for CMDs is likely to contain many 'false negatives' but few 'false positives', as documented previously.[12]

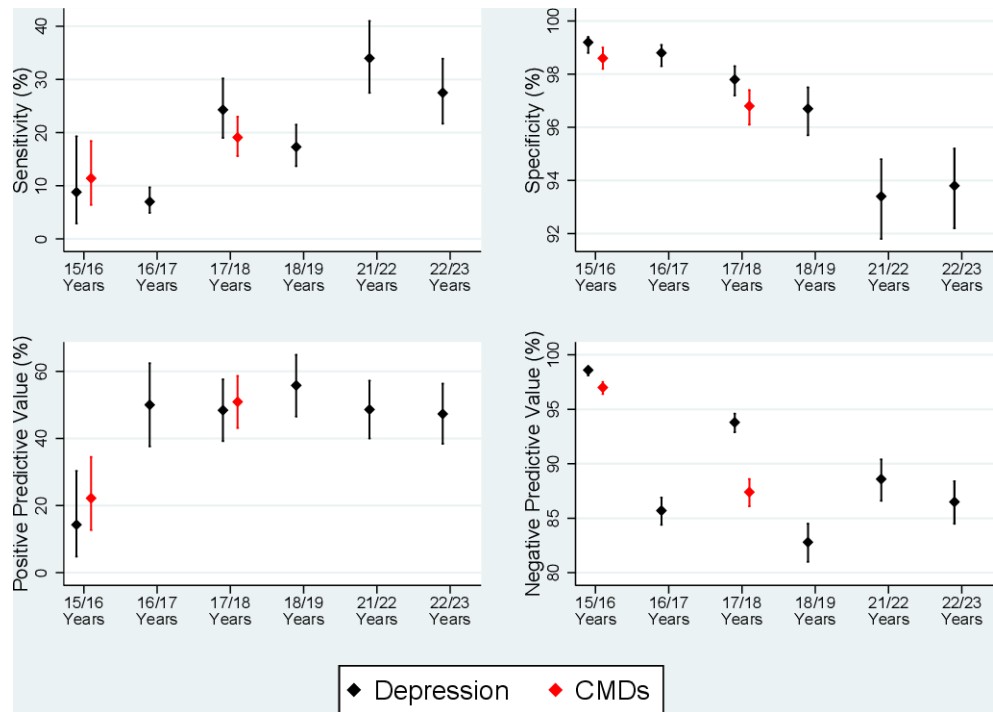

**Figure 2** Sensitivity, specificity and positive/negative predicted values for depression (black) and common mental disorders (CMDs; red) over each of the time points studies. Results are based on the definition 'current diagnosis, symptoms or treatment' to determine cases in primary care records, treating the Avon Longitudinal Study of Parents and Children data as the reference standard. Note that CMDs were only assessed at the age 15/16 and 17/18 clinics.

This study also explored the factors predicting identification of 'cases' (as identified in ALSPAC data) in primary care data. Consistent with previous research,[16 24] the strongest predictor was symptom severity, with individuals displaying more severe symptoms increasingly likely to be correctly classified. A history of CMDs, as well as increased rates of other mental health issues, somatic or general symptoms and engagement with primary care services (consultation and prescription rates), also predicted greater primary care identification rates. Many adolescents receive mental healthcare via specialised secondary care services, rather than through their GP, and this is reflected in referrals to secondary mental health services also being associated with higher identification rates. Unlike for the wider adult population, we

found little evidence that sociodemographic factors were consistently associated with case identification in primary care records for adolescents and young adults.[27 47]

Finally, this paper also presented a series of prediction models, which can be used by epidemiologists with access only to primary care data to predict CMDs in individuals who may not be formally diagnosed by a GP. Although the variance explained by these models is quite low, these models demonstrate that the inclusion of additional covariates from primary care records improved model fit, relative to models that contained only current diagnosis, symptoms or treatment. Out-of-sample prediction rates increased with age, suggesting that these models better predict depression/CMDs in young adulthood compared with adolescence. This is perhaps not surprising, given

**Table 5** In-sample and out-of-sample deviance ratios predicting depression and common mental disorders (CMDs) for each time point using all the available primary care data

| | Age 15/16 TF3 clinic (n=3663) | | Age 16/17 CCS quest. (n=3213) | Age 17/18 TF4 clinic (n=3084) | | Age 18/19 CCT quest. (n=1982) | Age 21/22 YPA quest. (n=1298) | Age 22/23 YPB quest. (n=1325) |
|---|---|---|---|---|---|---|---|---|
| | Dep. (%) | CMD (%) | Dep. (%) | Dep. (%) | CMD (%) | Dep. (%) | Dep. (%) | Dep. (%) |
| In-sample deviance ratio (60% training sample) | 9.9 | 8.4 | 8.3 | 14.6 | 9.2 | 9.0 | 11.7 | 13.4 |
| Out-of-sample deviance ratio (40% validation sample) | −1.3 | 2.9 | 4.3 | 7.8 | 7.8 | 8.0 | 12.6 | 9.1 |

Deviance ratios are taken from logistic cross-validation least absolute selection and shrinkage operator prediction models.

that rates of diagnoses from primary care data are low in adolescence and increase with age. However, comparison of the predicted sensitivities and specificities from these prediction models indicates that the improvement in detection of depression/CMDs relative to the primary care record data based on diagnosis, symptoms and treatment is minimal. We also acknowledge that these prediction models were only validated internally using ALSPAC data; before being used more widely, these models should be calibrated and validated externally using independent datasets from different populations (see, eg, Perry et al[48]).

### Strengths and limitations

A strength of this study is that it uses established methods to systematically define CMDs from primary care records,[10 11] allowing cross-study comparisons. This study uses a larger sample than a previous study using ALSPAC adolescent data,[10] and extends the age range assessed to adolescence through to early adulthood. This permits a broader view of how both ALSPAC-derived and primary care-derived CMD rates change with age, how sensitivities and specificities vary over the transition to adulthood, and how prediction models alter over this developmental stage. A further strength is that this study also uses a large, deeply phenotyped cohort, with depression and CMD measured at multiple time points using validated instruments.

This study has several limitations. The primary care data coding used may miss crucial information: possible diagnoses and symptoms may be noted within the 'free text' of routine electronic records,[12] which are generally not available for research purposes.[49] The primary care data only records pharmacological treatments prescribed by the GP, rather than psychological treatments provided by secondary care services. As the first line of treatment for adolescents is often psychological therapies, especially for mild depression[50], this may partially explain the lower sensitivities at younger ages. Although we included CAMHS referrals in our identification analyses and prediction models, this is still likely to underestimate the true prevalence of adolescent CMDs as only around one-third of children with a mental health disorder are referred to CAMHS.[23] Further, fewer than half of referrals to CAMHS in the UK are from a GP,[51] with school nurses, self-referrals and other routes to CAMHS possible. It is important to reinforce here that the aim of this paper was to detect and predict participants with CMDs using just primary care data, not to identify all potential individuals with a CMD from routine health records. For instance, it is possible that some participants may be 'detected' as having a CMD and referred to secondary healthcare services (eg, CAMHS) without having a CMD diagnosis in their primary care records if they did not present to their GP and were referred to secondary care services by another route. As we were interested in using just primary care records, these individuals would not appear as having a CMD in our data. Implications of this, and methods of improving detection rates when using routine health

records, are discussed in the Implications and recommendations section.

It is also possible that we have omitted other clinically relevant conditions noted in primary care record Read codes, such as additional personality disorders (eg, schizoid, borderline, obsessive-compulsive, etc), which are often associated with CMDs and may have improved the accuracy of our prediction models.[52] However, personality disorders are frequently difficult to diagnose in primary care settings,[53] and hence often not diagnosed by GPs.[54] Additionally, many personality disorders or proxies thereof—such as conduct disorders, eating disorders, psychosis, schizophrenia, self-harm, frequency of consultations, referral to secondary health services and CMDs themselves[52–55]—were included in the present study. Furthermore, these personality disorders were not consistent predictors of ALSPAC-derived CMDs (online supplemental table 13). Additionally, for all participants over the whole period covered in the study, there were approximately 14 500 Read codes relating to diagnoses of a mental health disorder, of which ~8000 were for CMDs and a further ~4200 were captured by other measured conditions (ADHD, eating disorders, conduct disorder, etc). In total, there were only 220 diagnosis codes for either personality disorder or obsessive-compulsive disorder, which related to 42 individuals. Omitting these additional personality disorders is therefore unlikely to substantially impact our conclusions or improve our prediction models. While this may be an important consideration if personality disorders were more commonly identified in the primary care records, we do not believe that this is a major limitation of this research.

A second limitation is that the linkage is primarily Bristol-based. As the cohort reaches adulthood they are more likely to move away from Bristol; as such, the proportion with linkage data drops from approximately two-thirds before age 18 to roughly one-third after this age. In addition to the resulting loss of statistical power and precision, there is also the potential for selection bias if those with linkage data systematically differ from those without.[56 57] At each time point, of those with ALSPAC data there are differences between those with and without linked primary care data in terms of SEP (eg, higher maternal education levels are associated with lower probability of having linkage data). Although differences in ALSPAC-derived CMDs appear minimal comparing those with vs without primary care data (table 3), it is possible that primary care data may differ between these groups. This may limit the generalisability of our prediction models; for example, compared with the whole ALSPAC cohort our sample with primary care data is biased towards those with a lower SEP, who may be less likely to attend GP appointments.[58] However, as in the wider ALSPAC cohort respondents tend to be from higher socioeconomic strata,[34] the impact of linkage data biased towards lower-SEP individuals on generalisability is uncertain. Comparing the primary care-derived CMD status of those with and without ALSPAC data we

observe few differences in terms of depression or CMDs at younger ages but, in adulthood, CMDs (although less so for depression) appear more prevalent among those with ALSPAC data (figure 1). One possible interpretation of this is that it reflects the demographics of ALSPAC respondents, as being female is associated with continued ALSPAC participation,[34] and females are at greater risk of CMDs.[59 60] When adjusting for sex these effects were somewhat attenuated, although participants with ALSPAC data at the 21/22 and 22/23 questionnaire time points were still more likely to have a primary care-derived CMD (online supplemental table 17). Inclusion of parental education (a proxy for SEP), which may also predict both continued ALSPAC participation and mental health, did not further diminish this effect. The selection pressures associated with having continued primary care linkage data in ALSPAC are likely to be complex and require further investigation to assess the potential for selection bias when using this data.

A related limitation is that as the research is specifically Bristol-based, generalisability to other populations, both in the UK and elsewhere, should be made with caution. For instance, the ALSPAC cohort is not representative of the UK national population, as ALSPAC contains a greater proportion of white and higher SEP individuals,[34] which is likely to shape health-seeking behaviour and GP engagement rates.[26 27] A further issue regarding generalisability is that the data in adolescence was collected between 2006 and 2011. Given the large shift in societal values towards increased visibility, awareness and understanding of mental health issues over the past few years, this may impact both GP decision-making and adolescents' health-seeking behaviour, potentially affecting diagnosis rates in this age group. Additional research is necessary to explore this among existing adolescents. As such, these models should be calibrated before use in other areas or calendar times.

A third limitation is the small numbers of individuals with CMD/depression in ALSPAC, especially at younger ages (and particularly the age 15/16 clinic data). This may explain why we failed to detect consistent sociodemographic differences in case identification, contrary to previous research with larger samples.[16 27 47] Larger studies are required to explore sociodemographic associations with identification in primary care records in greater detail, which—if present—are likely to be weaker than the effects of symptom severity.[16 24] In addition to the relatively small sample size, one potential reason for the lack of SEP gradient could be that SEP is based on parental SEP at the time of the study child's birth. Although parental SEP and child health outcomes are frequently correlated, this association is strongest in early childhood and tends to weaken with age.[26] Assessing the individual's SEP directly, particularly in early adulthood, may reveal these health inequalities. A further consequence of this paucity of diagnoses in adolescence is that we were not able to explore CMDs in early adolescence, despite many CMDs having their onset prior to age 15

years[5]; extending and replicating these results in a larger study which includes early adolescents would be an interesting avenue for future research.

A further limitation is that we have taken the ALSPAC data as the 'reference standard'. These measures may overdiagnose the presence of CMDs, especially in 'borderline' cases with less severe symptoms who may not visit their GP, thus increasing the number of false positives in the ALSPAC data. Although all of the instruments used in ALSPAC have been validated and are routinely used to screen for depression and CMDs,[38 39 41 44] previous studies have demonstrated that these questionnaire-based tools can provide quite divergent diagnoses of mental health conditions compared with standard clinical interviews (eg, CIS-R[40]). Additionally, apparent false negatives may also appear in the ALSPAC data if individuals are successfully receiving treatment to alleviate their CMD symptoms; in these cases, individuals would be diagnosed as having CMD via primary care records, but not via ALSPAC data.

### Implications and recommendations

Consistent with previous research,[12] this study has demonstrated that the rate of false negatives for CMDs in adolescents and young adults in routine primary care data is high. Thus, additional sources of information need to be used when working with routine health data. As fewer than half of referrals to CAMHS are from GPs,[51] using linkage data from CAMHS and other secondary mental healthcare services would likely increase detection rates. This would appear particularly important for adolescents, as the sensitivities at this age are much lower than in early adulthood. However, as CAMHS is over-subscribed, often only severe cases are accepted, potentially biasing these sources towards those with more severe CMD symptoms. Additionally, even in early adulthood sensitivities are still rather low (maximum 34% at age 21/22), suggesting that additional information is required to correctly identify CMDs in linkage data. One potential source of information is from the free-text fields in primary care records, which are not usually made available for research purposes.[12] However, although evidence suggests that using free text data can improve detection of medical conditions more generally,[61] the current evidence for CMDs—although limited to a small number of studies—suggests their inclusion only marginally improves detection rates.[12] Therefore, in addition to making use of more data sources, we also need better case detection at the primary care level to identify these 'invisible' cases who have a CMD but are not currently recorded as such in primary care records. As more severe CMD cases are more likely to be detected in primary care records, these missing cases are likely to present with milder CMD symptoms. Methods to improve GP detection of CMDs include: increasing GP confidence when identifying child and adolescent CMDs[22]; additional support and screening for at-risk groups (eg, after adverse life events), particularly in childhood and adolescence, to aid early detection of CMDs, as per best-practice guidelines[50 62]; and further training and resources for GPs

to aid identification and management of mental health issues.[20 21]

## CONCLUSION

We have demonstrated how routine electronic primary care data can be used with cohort study data to estimate the size of the 'clinical iceberg' of undetected CMDs in primary care data throughout adolescence and early adulthood, and to describe the characteristics of those less likely to be identified as cases in primary care records. Although overall sensitivities were low, both sources of data accurately predicted individuals with more severe CMD symptoms. The number of individuals diagnosed as having a CMD, and the correspondence between ALSPAC and primary care data, increased with age. Additional sources of data—for example, from secondary care services such as CAMHS, or from free text fields—might be required to determine CMD prevalence more accurately, particularly in adolescence. Development of further prediction models may improve estimation of prevalence of CMDs from primary care records and help target interventions to individuals with CMDs who would otherwise not be identified as cases in primary care records. This should be conducted in tandem with methods to improve case detection of CMDs among adolescents and young adults by primary care clinicians.

**Contributors** RPC, KT, SLP, KW, GS and JD conceived and designed the study. RPC processed the primary care data. DS performed the statistical analyses and drafted the manuscript. RPC, KT, SLP, KW, GS and JD contributed to the interpretation of the results. All authors commented on the draft, and have read and approved the final version of the manuscript. DS will serve as guarantor for the contents of this paper.

**Funding** This work was funded by the Medical Research Council (MRC grant number: MC_PC_17210). The UK Medical Research Council and Wellcome (Grant ref: 217065/Z/19/Z) and the University of Bristol provide core support for ALSPAC. A comprehensive list of grants funding is available on the ALSPAC website (http://www.bristol.ac.uk/alspac/external/documents/grant-acknowledgements.pdf http://www.bristol.ac.uk/alspac/external/documents/grant-acknowledgements.pdf); collection of ALSPAC CMD data was funded by the NIH (grant references 5R01MH073842-04 and PD301198-SC101645; for the TF3 DAWBA and YPB MFQ data), Wellcome Trust (grant reference 08426812/Z/07/Z; for the TF4 CIS-R data), and a joint Wellcome Trust and MRC grant (grant reference 092731; for the CCS, CCT and YPA SMFQ data). This publication is the work of the authors and Daniel Smith and Rosie Cornish will serve as guarantors for the contents of this paper.

**Competing interests** None declared.

**Patient consent for publication** Not applicable.

**Ethics approval** Ethical approval for the study was obtained from the ALSPAC Ethics and Law Committee and the Local Research Ethics Committees (NHS Haydock REC: 10/H1010/70). Informed consent for the use of data collected via questionnaires and clinics was obtained from participants following the recommendations of the ALSPAC Ethics and Law Committee at the time.

**Provenance and peer review** Not commissioned; externally peer reviewed.

**Data availability statement** ALSPAC data access is through a system of managed open access. Information about access to ALSPAC data is given on the ALSPAC website (http://www.bristol.ac.uk/alspac/researchers/access/).

**ORCID iDs**
Daniel Smith http://orcid.org/0000-0001-6467-2023
Kathryn Willan http://orcid.org/0000-0002-1566-6054
Stephanie L Prady http://orcid.org/0000-0002-8933-8045
Josie Dickerson http://orcid.org/0000-0003-0121-3406
Gillian Santorelli http://orcid.org/0000-0003-0427-1783
Rosie Peggy Cornish http://orcid.org/0000-0002-2874-7646

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
