## [Reviewer comments · BMJ Open]

ARTICLE DETAILS

TITLE (PROVISIONAL)	Assessing and predicting adolescent and early adulthood common mental disorders using electronic primary care data: analysis of a prospective cohort study (ALSPAC) in southwest England
AUTHORS	Smith, Daniel; Willan, Kathryn; Prady, Stephanie; Dickerson, Josie; Santorelli, Gillian; Tilling, Kate; Cornish, Rosie

VERSION 1 – REVIEW

REVIEWER	Osimo, Emanuele Cambridge University, Psychiatry
REVIEW RETURNED	15-Jul-2021

GENERAL COMMENTS	Thanks for asking me to review this manuscript, which, with a few exceptions, is exceptionally clear and well written. The Authors set out to compare primary care data against validated measures of CMDs at multiple time points during adolescence and young adulthood from the ALSPAC birth cohort. They aim to replicate a previous study with a much smaller sample, aiming to explore agreement between primary care records and cohort data over time. They also aim to assess the factors impacting rates of identification in primary care records. Finally, they build prediction models using outcomes from ALSPAC, and predictors from primary care data. I think that the first 2 objectives are very successful. The 3rd objective I find a bit less strong, as the study team does not have an external validation sample and does not present measures of calibration. Overall, the abstract is not very clear, and should be re-written if possible. The corresponding parts of the introduction, methods and results are much clearer and more accessible. There are a few more specific comments below, but with some clarifications and specifying the limited generalisability of the prediction models, I believe this to be important work to publish with relatively minor edits. Also, I found the Authors have taken very good care of clearly stating the limitations of their work, which is commendable. 1) ABSTRACT: 1a) I believe the abstract could be clearer on the prediction side of things. The Authors state: "3) taking ALSPAC as the reference standard, to construct models predicting ALSPAC-derived CMDs using primary care data.". From first reading this, it might be unclear what the Authors are actually trying to predict/validate. Are the Authors trying to validate the predictive accuracy of GP records, using ALSPAC-based diagnoses as gold standard?
---

1b) The same can be said for the "Outcome measures" section below, where the Authors state: "Lasso models were then performed to predict ALSPAC CMDs from primary care data.". Unclear what is the testing/validation sample, and what is being predicted.

1c) "Results" section: "Sensitivities were low for CMDs (range: 3.5 to 19.1%) and depression (range: 1.6 to 34.0%), while specificities were high (nearly all >95%). The strongest predictor of identification in the primary care data was symptom severity." Sensitivity for what and compared to what? Strongest predictor of identification of what? Symptom severity in ALSPAC or GP records?

1d) the objectives as stated in the Abstract don't match those on page 5, lines 12-19 of the manuscript (the second being much clearer.)

1e) I believe the Abstract should contain more fine detail about the LASSO prediction modelling performed. I believe it should summarise information from your methods: "LASSO models ... were used at each time point to assess the combination of variables from primary care data which best predicted depression/CMDs from the ALSPAC data." Also it should mention that you only internally validate your findings by splitting your sample.

Page 3, Strengths and limitations of this study: very clear and informative.

Page 5, line 29: typo "AS total" instead of "A total".

2) RESULTS:

2a) these are generally very clear, but can be a bit technical. For example, on page 13 lines 44 and subsequent, "in-sample deviance ratios are quite low (8.3 to 14.6%). Out-of-sample deviance ratios are lower (-1.3 to 12.6%) but do increase with age". These numbers do not necessarily mean much to the general reader, and while you explain deviance ratios on line 53 of page 8, I could not find an explanation for in-sample or out-of-sample ones.

3) DISCUSSION:

3a) "This suggests that the primary care data is likely to contain many 'false negatives' but few 'false positives'". Does this apply only narrowly to depression and anxiety diagnoses? If so, please specify.

3b) And, on a related note, do you have any data/evidence on personality disorders? They are probably, when on the mild side, one of the most prevalent MH presentations to primary care, and they are hardly ever diagnosed by GPs. I believe you should discuss the presence (if you have any) or absence of data as a limitation if not.

3c) "Finally, this paper also presented a series of prediction models, which can be used by epidemiologists with access only to primary care data to predict CMDs in individuals who may not be formally diagnosed by a GP." While it is commendable that the Authors have published and shared the full models in S13 and S14 for others to replicate their findings, which is rare, they should probably more prominently acknowledge that they have performed internal development and internal validation, but not external validation. They have neither shown how well the model calibrate.

	Therefore, they should probably say that their models, before being of use, should be externally validated.
--	---

REVIEWER	van der westhuizen, Claire
REVIEW RETURNED	24-Jul-2021

GENERAL COMMENTS	Thank you for the opportunity to review this paper. The strengths of the study include: the data is sourced from a prospective cohort study, with links to primary care data and careful description of methods for primary study and this study. I recommend some revisions which I think will clarify aspects and also add to the literature in the global mental health field. To my mind, the major overall gap is that the relevance of the study findings for the clinical setting is not discussed. The data could be helpful for healthcare policymakers and planners, but the relevance for clinicians and patients is not clear. In my opinion, the addition of some points regarding clinical practice would help contextualise the findings and make this paper interesting/relevant for a broader readership. Abstract In the abstract, it would be helpful to state the proportion of participants with both primary care and ALSPAC data; the frequencies for 2 age groups are not that informative. Introduction 1. In the introduction reasons for CMDs being missed in primary care are listed, but omit two common reasons, namely that: (i) individuals with mental health difficulties, particularly young people, often do not articulate their mental health symptoms, but may present with somatic complaints, or other unrelated complaints but do not mention their other problems (this is often related to low mental health literacy and help-seeking behaviour in this group); and (ii) GP visit slots are often far too brief to elicit sensitive issues, such as mental health complaints. (See 3 papers which could be relevant: https://pubmed.ncbi.nlm.nih.gov/31015266/, https://pubmed.ncbi.nlm.nih.gov/14505065/ and https://www.ncbi.nlm.nih.gov/pmc/articles/PMC5033306/) 2. The authors mention a UK study that found that 'non-British ethnicity ... predicted lower rates of CMD detection'. I am assuming that the authors meant 'nationality' as being British is not associated with a certain ethnicity to the exclusion of other ethnicities (?). There is literature regarding the differences in healthcare services provided to people of minority groups – regarding immigration status, ethnicity (white vs people of colour) etc. Methods 1. In the last paragraph on page 5, participants enrolled included all 'singletons and twins'. Is there a reason for mentioning this specifically: were any triplets excluded? Why not just 14 731 participants enrolled as children? 2. Records were included from the age of 15 years. Is there a reason that younger adolescents were not included (10-14 years)? Around 50% of mental disorders have their onset by the age of 14 years, so these data would be important to include. 3. Under the ALSPAC data heading on page 6, there is a mention of the 'estimated probability' guiding the depression diagnosis. Many diagnostic tools identify the presence of a disorder or not based on ICD or DSM criteria. How is the probability calculated for the DAWBA?
---

	4. In the 2nd paragraph under this heading, there is mention that the criteria of 'mild depression' included 'moderate and severe depression' which I find very confusing. Please clarify. 5. It would be helpful to describe 'Read' codes briefly for international readers. Discussion The authors mention improving 'detection' by using CAMHS data etc, although these cases are already detected – not in the data, but by clinicians who have detected symptoms indicating a CMD diagnosis or risk for developing a disorder, and have referred the patient. Mention of improving case detection would surely bring up the 'mental health screening in primary care' debate (https://pediatrics.aappublications.org/content/141/3/e20174081.long; https://www.nice.org.uk/guidance/ng134/chapter/Recommendations#step-1-detection-risk-profiling-and-referral). Are primary care healthcare staff using the guidelines and screening young people after an 'undesirable life event' etc (implications for further research)? Given the limitations of primary care data and the lack of sufficient predictors of CMD diagnoses, does this not suggest that improved data solutions or case detection is needed for early intervention for mental health (as well as for those planning services)? Globally, mental health difficulties, particularly in children and adolescents, are under-diagnosed and young people often do not have access to effective treatment.
--	---

VERSION 1 – AUTHOR RESPONSE

Reviewer: 1

Dr. Emanuele Osimo, Cambridge University

Comments to the Author:

Thanks for asking me to review this manuscript, which, with a few exceptions, is exceptionally clear and well written. The Authors set out to compare primary care data against validated measures of CMDs at multiple time points during adolescence and young adulthood from the ALSPAC birth cohort. They aim to replicate a previous study with a much smaller sample, aiming to explore agreement between primary care records and cohort data over time. They also aim to assess the factors impacting rates of identification in primary care records. Finally, they build prediction models using outcomes from ALSPAC, and predictors from primary care data.

I think that the first 2 objectives are very successful. The 3rd objective I find a bit less strong, as the study team does not have an external validation sample and does not present measures of calibration.

Overall, the abstract is not very clear, and should be re-written if possible. The corresponding parts of the introduction, methods and results are much clearer and more accessible.

There are a few more specific comments below, but with some clarifications and specifying the limited generalisability of the prediction models, I believe this to be important work to publish with relatively minor edits.

Also, I found the Authors have taken very good care of clearly stating the limitations of their work, which is commendable.

We are delighted with the reviewer's largely positive review and thank the reviewer for their constructive comments. We hope that we have addressed all their concerns in our responses below, especially regarding clarity in the abstract. Our responses to the reviewer's comments are in blue, below the original comments.

1) ABSTRACT:

1a) I believe the abstract could be clearer on the prediction side of things.

The Authors state: "3) taking ALSPAC as the reference standard, to construct models predicting ALSPAC-derived CMDs using primary care data.". From first reading this, it might be unclear what the Authors are actually trying to predict/validate. Are the Authors trying to validate the predictive accuracy of GP records, using ALSPAC-based diagnoses as gold standard?

Our aim for objective 3 was not to validate the predictive accuracy of primary care records against ALSPAC-based diagnoses (this was objective 1), but rather to use as much primary care data as possible to predict ALSPAC-based CMD diagnoses. This may be useful for researchers to predict CMDs, including those potentially missing CMD diagnoses in these primary care records, using only primary care records without access to 'gold standard' CMD diagnoses from validated questionnaires. We have tried to make this clearer by changing objective 3 in the abstract to "3) construct models predicting ALSPAC-derived CMDs using only primary care data". However, due to space constraints on the abstract (max 300 words) we are unable to provide a more detailed description here. We hope that this makes our objective clearer, and readers can find more detailed information in the introduction.

1b) The same can be said for the "Outcome measures" section below, where the Authors state: "Lasso models were then performed to predict ALSPAC CMDs from primary care data.". Unclear what is the testing/validation sample, and what is being predicted.

We have changed this to "Lasso models were used at each time point to predict ALSPAC-derived CMDs using only primary care data, with internal validation by randomly splitting data into 60% training and 40% validation samples" to specify the testing/validation sample and make it clearer that we were predicting ALSPAC CMDs (from validated questionnaires) using only primary care data.

1c) "Results" section: "Sensitivities were low for CMDs (range: 3.5 to 19.1%) and depression (range: 1.6 to 34.0%), while specificities were high (nearly all >95%). The strongest predictor of identification in the primary care data was symptom severity." Sensitivity for what and compared to what? Strongest predictor of identification of what? Symptom severity in ALSPAC or GP records? Definitions of sensitivities/specificities are given in the 'Outcome measures' section of the abstract (and have been updated slightly to improve clarity): "For each time point, sensitivities and specificities for primary care CMD diagnoses were calculated for predicting ALSPAC-derived measures of CMDs...". To help make this clearer in the 'Results' section of the abstract, we have updated the text to (new text in italics): "Sensitivities for primary care diagnoses were low for CMDs (range: 3.5 to 19.1%) and depression (range: 1.6 to 34.0%), while specificities were high (nearly all >95%)". Similarly, regarding 'identification' we have updated this text for clarity, both in terms of what is being identified (i.e., whether participants with ALSPAC-derived CMD are identified in the primary care data) and details of symptom severity (i.e., symptom severity in both ALSPAC and primary care records were strong predictors of CMD identification in the primary care records - this is detailed in the main text, but we have not added this additional information into the abstract to avoid going over the word limit). To make both of these points clearer, we have updated this sentence to: "The strongest predictors of identification in the primary care data for those with ALSPAC-derived CMDs were symptom severity indices."

1d) the objectives as stated in the Abstract don't match those on page 5, lines 12-19 of the manuscript (the second being much clearer.)

We do not quite understand the reviewer here. Objectives 2 and 3 are the same, although with slightly different wording to save space in the abstract. Objective 1 is almost the same, but with the "replicate and expand the results of a previous ALSPAC study" part removed from the abstract to keep within the abstract word limit. We have added in the phrase "over adolescence and young adulthood" to objective 1 in the abstract, however, to specify the age range under investigation: "1) examine agreement between common mental disorders (CMDs) from primary care records and repeated CMD

questionnaire data from ALSPAC (the Avon Longitudinal Study of Parents and Children) over adolescence and young adulthood”.

1e) I believe the Abstract should contain more fine detail about the LASSO prediction modelling performed. I believe it should summarise information from your methods: "LASSO models ... were used at each time point to assess the combination of variables from primary care data which best predicted depression/CMDs from the ALSPAC data." Also it should mention that you only internally validate your findings by splitting your sample.

We have provided additional information about internal validation (see response to point 1b, above), and have also added additional information that these lasso models were performed at each time point. We believe that this summarises the key information necessary for readers regarding the lasso prediction models and is sufficient detail for readers to understand the broad aims of these models; given the tight word limit and the need to describe the other objectives/results in sufficient detail, we do not feel that we can include any additional fine detail about the lasso prediction modelling in the abstract. Interested readers can find this more detailed description in the main body of the manuscript.

However, if the editor feels that this information is necessary, we would be happy to include this. This would obviously take us over 300 words.

Page 3, Strengths and limitations of this study: very clear and informative.

Page 5, line 29: typo "AS total" instead of "A total".
This has now been updated.

2) RESULTS:

2a) these are generally very clear, but can be a bit technical. For example, on page 13 lines 44 and subsequent, "in-sample deviance ratios are quite low (8.3 to 14.6%). Out-of-sample deviance ratios are lower (-1.3 to 12.6%) but do increase with age". These numbers do not necessarily mean much to the general reader, and while you explain deviance ratios on line 53 of page 8, I could not find an explanation for in-sample or out-of-sample ones.

When describing the methods on page 8, to assist readers we have provided additional detail about the interpretation of deviance ratios and clarified that the in-sample deviance ratios refer to the 60% training sample and that the out-of-sample deviance ratios refer to the 40% validation sample: "We randomly split our sample into 60% training and 40% validation samples, and then compared the deviance ratios for each to inspect how well the training model performed when predicting depression/CMDs in the validation sample. The deviance ratio is a goodness-of-fit statistic generally between 0 and 1 comparable to R^2 , but for non-linear models, indicating the fraction of deviance explained relative to the null constant-only model (with values closer to 1 indicating better model fit; see page 33 of [45]). In-sample deviance ratios refer to results from the 60% training sample, while out-of-sample deviance ratios refer to results from the 40% validation sample (note that as the coefficients derived from the training sample are used in the validation sample, it is possible to observe negative deviance ratios, indicating that the coefficients are worse at predicting the outcome than the null model)."

3) DISCUSSION:

3a) "This suggests that the primary care data is likely to contain many 'false negatives' but few 'false positives'". Does this apply only narrowly to depression and anxiety diagnoses? If so, please specify. This applies specifically to common mental disorders, and have amended this sentence accordingly: "This suggests that the primary care data for CMDs is likely to contain many 'false negatives' but few 'false positives', as documented previously".

3b) And, on a related note, do you have any data/evidence on personality disorders? They are probably, when on the mild side, one of the most prevalent MH presentations to primary care, and they are hardly ever diagnosed by GPs. I believe you should discuss the presence (if you have any) or absence of data as a limitation if not.

A number of mental health conditions or associated manifestations/comorbidities, including personality disorders, were included in the Read code list for the primary care data (table S1), including conduct disorders, psychosis, schizophrenia, eating disorder, bipolar disorder, self-harm, substance abuse, autism spectrum disorder, ADHD, referral to secondary mental health services and CMDs themselves. Nonetheless, many personality disorders – such as schizoid, schizotypal, borderline, histrionic, narcissistic, avoidant, dependent and obsessive-compulsive disorders – were not included in our analyses.

We have included this as a potential limitation in the discussion, although we believe that this is unlikely to impact our conclusions as the number of these other personality disorders in the primary care data was very low: “It is also possible that we have omitted other clinically-relevant conditions noted in primary care record Read codes, such as additional personality disorders (e.g., schizoid, borderline, obsessive-compulsive, etc.), which are often associated with CMDs and may have improved the accuracy of our prediction models [52]. However, personality disorders are frequently difficult to diagnose in primary care settings [53], and hence often not diagnosed by GPs [54].

Additionally, many personality disorders, or proxies thereof – such as conduct disorders, eating disorders, psychosis, schizophrenia, self-harm, frequency of consultations, referral to secondary health services and CMDs themselves [52–55] – were included in the present study, yet were not consistent predictors of ALSPAC-derived CMDs (Table S13). Additionally, for all participants over the whole period covered in the study, there were approximately 14,500 Read codes relating to diagnoses of a mental health disorder, of which ~8,000 were for CMDs and a further ~4,200 were captured by other measured conditions (ADHD, eating disorders, conduct disorder, etc.). In total, there were only 220 diagnosis codes for either personality disorder or obsessive-compulsive disorder, which related to 42 individuals. Omitting these additional personality disorders is therefore unlikely to substantially impact our conclusions or improve our prediction models. While this may be an important consideration if personality disorders were more commonly-identified in the primary care records, we do not believe that this is a major limitation of this research.”

3c) "Finally, this paper also presented a series of prediction models, which can be used by epidemiologists with access only to primary care data to predict CMDs in individuals who may not be formally diagnosed by a GP." While it is commendable that the Authors have published and shared the full models in S13 and S14 for others to replicate their findings, which is rare, they should probably more prominently acknowledge that they have performed internal development and internal validation, but not external validation. They have neither shown how well the model calibrate. Therefore, they should probably say that their models, before being of use, should be externally validated.

We agree that this is an important point, and thanks for raising it. We have now included an additional sentence in the discussion on this: “We also acknowledge that these prediction models were only validated internally using ALSPAC data; before being used more widely, these models should be calibrated and validated externally using independent datasets from different populations (see, e.g., [46]).”

Reviewer: 2

Dr. Claire van der westhuizen

Comments to the Author:

Thank you for the opportunity to review this paper. The strengths of the study include: the data is sourced from a prospective cohort study, with links to primary care data and careful description of methods for primary study and this study. I recommend some revisions which I think will clarify aspects and also add to the literature in the global mental health field.

We thank the reviewer for their comments which have strengthened the manuscript, and hope that we have addressed all their concerns in our responses below. Our responses to the reviewer's comments are in blue, below the original comments.

To my mind, the major overall gap is that the relevance of the study findings for the clinical setting is not discussed. The data could be helpful for healthcare policymakers and planners, but the relevance for clinicians and patients is not clear. In my opinion, the addition of some points regarding clinical practice would help contextualise the findings and make this paper interesting/relevant for a broader readership.

We thank the reviewer for their helpful suggestions, and have now updated the manuscript in several places, including:

The 'Conclusion' section of the abstract has been rewritten: "Primary care data underestimate CMDs compared to population-based studies. Improving GP identification, and using free-text data or secondary care data, is needed to improve the accuracy of models using clinical data."

End of the 'implications and recommendations' section in discussion: "Therefore, in addition to making use of more data sources, we also need better case detection at the primary care level to identify these "invisible" cases who have a CMD but are not currently recorded as such in primary care records. As more severe CMD cases are more likely to be detected in primary care records, these missing cases are likely to present with milder CMD symptoms. Methods to improve GP detection of CMDs include: increasing GP confidence when identifying child and adolescent CMDs [22]; additional support and screening for at-risk groups (e.g., after adverse life events), particularly in childhood and adolescence, to aid early detection of CMDs, as per best-practice guidelines [50,62]; and further training and resources for GPs to aid identification and management of mental health issues [20,21]."

End of 'conclusion' section: "Development of further prediction models may improve estimation of prevalence of CMDs from primary care records and help target interventions to individuals with CMDs who would otherwise not be identified as cases in primary care records. This should be conducted in tandem with methods to improve case detection of CMDs among adolescents and young adults by primary care clinicians."

Abstract

In the abstract, it would be helpful to state the proportion of participants with both primary care and ALSPAC data; the frequencies for 2 age groups are not that informative.

This has now been updated: "Between 31% (3,633; age 15/16) and 11% (1,298; age 21/22) of participants had both primary care and ALSPAC CMD data."

Introduction

1. In the introduction reasons for CMDs being missed in primary care are listed, but omit two common reasons, namely that: (i) individuals with mental health difficulties, particularly young people, often do not articulate their mental health symptoms, but may present with somatic complaints, or other unrelated complaints but do not mention their other problems (this is often related to low mental health literacy and help-seeking behaviour in this group); and (ii) GP visit slots are often far too brief to elicit sensitive issues, such as mental health complaints. (See 3 papers which could be relevant: <https://pubmed.ncbi.nlm.nih.gov/31015266/>, <https://pubmed.ncbi.nlm.nih.gov/14505065/> and <https://www.ncbi.nlm.nih.gov/pmc/articles/PMC5033306/>)

Thank you for these suggestions and for the references. These have now been added to the introduction: "Additionally, GPs may fail to identify, or be less willing to diagnose CMDs or prescribe

medication to these groups, potentially in part due to a lack of confidence of GPs in identifying and managing CMDs in children and adolescents, low mental health literacy and help-seeking behaviour among these groups, and GP visits being too brief to elicit such sensitive information [18–22].”.

2. The authors mention a UK study that found that ‘non-British ethnicity ... predicted lower rates of CMD detection’. I am assuming that the authors meant ‘nationality’ as being British is not associated with a certain ethnicity to the exclusion of other ethnicities (?). There is literature regarding the differences in healthcare services provided to people of minority groups – regarding immigration status, ethnicity (white vs people of colour) etc. This should have been ‘ethnicities other than White British’, and has now been updated accordingly.

Methods

1. In the last paragraph on page 5, participants enrolled included all ‘singletons and twins’. Is there a reason for mentioning this specifically: were any triplets excluded? Why not just 14 731 participants enrolled as children?

Yes, enrolled children from triplet or quadruplet births are automatically excluded from all ALSPAC datasets. We have added a short statement in parentheses to explain this: “(data from triplet/quadruplet births were excluded due to confidentiality reasons, as per standard ALSPAC protocols)”.

2. Records were included from the age of 15 years. Is there a reason that younger adolescents were not included (10-14 years)? Around 50% of mental disorders have their onset by the age of 14 years, so these data would be important to include.

We chose the age of 15 years as our first time point because this was the first age at which common mental disorders (depression and anxiety) were measured together, using the DAWBA questionnaire. Additionally, given the paucity of primary care CMD diagnoses at age 15 (only ~0.2% of the sample; Figure 1) and the issues with analyses of the data at age 15 (lack of power due to so few participants having a primary care depression/CMD diagnosis), there would be little benefit of extending these analyses to younger adolescents in the ALSPAC sample where these issues are likely to be exacerbated.

However, we do agree that extending and replicating these results in a larger study which includes younger adolescents would be very interesting and worthwhile, and have included a sentence on this as a limitation/area for future work in the discussion: “A further consequence of this paucity of diagnoses in adolescence is that we were not able to explore CMDs in early adolescence, despite many CMDs having their onset prior to age 15 years [5]; extending and replicating these results in a larger study which includes early adolescents would be an interesting avenue for future research.”.

3. Under the ALSPAC data heading on page 6, there is a mention of the ‘estimated probability’ guiding the depression diagnosis. Many diagnostic tools identify the presence of a disorder or not based on ICD or DSM criteria. How is the probability calculated for the DAWBA?

The raw ALSPAC data was based on DAWBA ‘bands’ of estimated probabilities of diagnosis, with an estimated probability of >50% for a diagnosis according to these bands identical to a diagnosis based on DSM-IV and ICD-10 criteria. We have updated this text to improve clarity: “At the age 15/16 clinic, depression and anxiety were assessed using the Development and Well-Being Assessment (DAWBA) interview [38], which identifies several psychiatric diagnoses in children and adolescents (based on International Classification of Diseases-10 (ICD-10) and Diagnostic and Statistical Manual of Mental Disorders fourth edition (DSM-IV) criteria). Here, in addition to a diagnosis for depression by itself, we defined CMDs as a diagnosis of depression and/or any anxiety disorder (generalised anxiety disorder, panic disorder, agoraphobia, social phobia and specific phobias).”.

4. In the 2nd paragraph under this heading, there is mention that the criteria of ‘mild depression’ included ‘moderate and severe depression’ which I find very confusing. Please clarify.

We wish to clarify that by defining ‘depression’ as ‘mild depression’, this was not to the exclusion of moderate or severe depression. In an attempt to make this clearer, we have updated this to: “Here, anyone diagnosed with mild, moderate or severe depression was classified as having depression, while a diagnosis of CMD was defined as having depression and/or an anxiety disorder...”.

5. It would be helpful to describe ‘Read’ codes briefly for international readers. We have added the following description to the ‘electronic primary care data’ section: “Read codes are a comprehensive list of standardised clinical terms used by healthcare professionals within the UK National Health Service to record clinical information (they have since been replaced by ‘SNOMED CT’ codes, but our data contained Read codes as they pre-dated this change).”.

Discussion

The authors mention improving ‘detection’ by using CAMHS data etc, although these cases are already detected – not in the data, but by clinicians who have detected symptoms indicating a CMD diagnosis or risk for developing a disorder, and have referred the patient.

If we understand this point correctly, we agree that where patients are part of CAHMS (or similar), these cases are already ‘detected’ in terms of being referred to/treated by secondary mental healthcare services. However, if one’s aim – as ours was here – is to detect and predict patients with CMDs using primary care data, then if these patients were not diagnosed or referred to secondary care services by their GP they would not appear in primary care records. This would seem especially likely amongst adolescents, as in this group fewer than half of referrals to CAMHS are from a GP (e.g., being via school nurse, self-referral, or other routes). As we discuss in the ‘Strengths and limitations’ and ‘Implications and recommendations’ sections of the discussion, inclusion of CAMHS and other secondary care data, as well as greater detection by GPs during primary care, would therefore likely improve agreement with the ALSPAC questionnaire data.

In an attempt to forestall other readers drawing similar conclusions, we have included the following in the second paragraph of the ‘strengths and limitations’ section of the discussion: “It is important to reinforce here that the aim of this paper was to detect and predict participants with CMDs using just primary care data, not to identify all potential individuals with a CMD from routine health records. For instance, it is possible that some participants may be ‘detected’ as having a CMD and referred to secondary healthcare services (e.g., CAMHS) without having a CMD diagnosis in their primary care records if they did not present to their GP and were referred to secondary care services by another route. As we were interested in using just primary care records, these individuals would not appear as having a CMD in our data. Implications of this, and methods of improving detection rates when using routine health records, are discussed in the ‘Implications and recommendations’ section below.”.

Mention of improving case detection would surely bring up the ‘mental health screening in primary care’ debate (<https://pediatrics.aappublications.org/content/141/3/e20174081.long>; <https://www.nice.org.uk/guidance/ng134/chapter/Recommendations#step-1-detection-riskprofiling-and-referral>). Are primary care healthcare staff using the guidelines and screening young people after an ‘undesirable life event’ etc (implications for further research)? Given the limitations of primary care data and the lack of sufficient predictors of CMD diagnoses, does this not suggest that improved data solutions or case detection is needed for early intervention for mental health (as well as for those planning services)? Globally, mental health difficulties, particularly in children and adolescents, are under-diagnosed and young people often do not have access to effective treatment.

This is an important point, which we hope we have addressed in our first point above concerning relevance for clinicians by including additional discussion about methods to improve GP detection rates via training, support and screening for at-risk groups (e.g., at the end of the ‘implications and recommendations’ section in discussion).

VERSION 2 – REVIEW

REVIEWER	Osimo, Emanuele Cambridge University, Psychiatry
REVIEW RETURNED	06-Sep-2021

GENERAL COMMENTS	Thanks for appropriately addressing all of my comments. The paper is now ready for publication.
--

REVIEWER	van der westhuizen, Claire
REVIEW RETURNED	14-Sep-2021

GENERAL COMMENTS	Thank you for the clarifications and the added details. The paper reads very well. Congrats!
--